# Real-World Biomarkers for Pediatric Takayasu Arteritis

**DOI:** 10.3390/ijms25137345

**Published:** 2024-07-04

**Authors:** Lieselot Peremans, Marinka Twilt, Susanne M. Benseler, Silviu Grisaru, Adam Kirton, Kimberly A. Myers, Lorraine Hamiwka

**Affiliations:** 1Section of Nephrology, Department of Pediatrics, Alberta Children’s Hospital, Cumming School of Medicine, University of Calgary, Calgary, AB T2N 1N4, Canada; 2Section of Rheumatology, Department of Pediatrics, Alberta Children’s Hospital, Cumming School of Medicine, University of Calgary, Calgary, AB T2N 1N4, Canada; marinka.twilt@albertahealthservices.ca (M.T.);; 3Children’s Health Ireland, D01 R5P3 Dublin, Ireland; 4Section of Neurology, Departments of Pediatrics and Clinical Neurosciences, Alberta Children’s Hospital, University of Calgary, Calgary, AB T2N 1N4, Canada; 5Section of Cardiology, Department of Pediatrics, Alberta Children’s Hospital, University of Calgary, Calgary, AB T2N 1N4, Canada

**Keywords:** Takayasu arteritis, childhood vasculitis, pediatrics, biomarkers, diagnosis, disease activity, review

## Abstract

Childhood-onset Takayasu arteritis (TA) is a rare, heterogeneous disease with limited diagnostic markers. Our objective was to identify and classify all candidates for biomarkers of TA diagnosis in children reported in the literature. A systematic literature review (PRISMA) of MEDLINE, EMBASE, Wiley Cochrane Library, ClinicalTrias.gov, and WHO ICTRP for articles related to TA in the pediatric age group between January 2000 and August 2023 was performed. Data on demographics, clinical features, laboratory measurements, diagnostic imaging, and genetic analysis were extracted. We identified 2026 potential articles, of which 52 studies (81% case series) met inclusion criteria. A total of 1067 TA patients were included with a peak onset between 10 and 15 years. Childhood-onset TA predominantly presented with cardiovascular, constitutional, and neurological symptoms. Laboratory parameters exhibited a low sensitivity and specificity. Imaging predominantly revealed involvement of the abdominal aorta and renal arteries, with magnetic resonance angiography (MRA) being the preferred imaging modality. Our review confirms the heterogeneous presentation of childhood-onset TA, posing significant challenges to recognition and timely diagnosis. Collaborative, multinational efforts are essential to better understand the natural course of childhood-onset TA and to identify accurate biomarkers to enhance diagnosis and disease management, ultimately improving patient outcomes.

## 1. Introduction

Takayasu arteritis (TA) is a granulomatous inflammatory large-vessel vasculitis that primarily affects the aorta and its major branches, as well as the coronary and pulmonary arteries. This inflammation, if left untreated, can lead to segmental stenosis, occlusion, dilatation, and/or aneurysms of the affected arteries. While TA is the most common form of large-vessel vasculitis in children, epidemiological data on childhood-onset TA are limited and vary based on geographical location [1,2]. The diagnosis of TA relies on the combined evaluation of clinical features, various imaging modalities, and laboratory findings [3]. Unfortunately, there is often a significant delay in diagnosing childhood-onset TA. Large-vessel vasculitis is a hidden disease manifesting as systemic features and organ dysfunction due to ischemia and/or inflammation. Health care providers may lack familiarity with its clinical presentation due to its rarity, and the disease commonly manifests with nonspecific systemic symptoms that resemble those of other, more prevalent conditions. Additionally, the variable localization and extent of the vessel inflammation can lead to significant variation in clinical symptoms. Nevertheless, improving disease outcomes depends on a rapid and accurate diagnosis as well as a careful monitoring of disease activity. The recognition of increased inflammatory disease activity may be mitigated by timely intervention with targeted immunosuppressive agents.

Patient outcomes are closely related to the duration and extent of vasculitis, making early recognition and diagnosis critical. In clinical practice, our aim is to select the optimal treatment for each patient, one that effectively induces and maintains disease remission while limiting adverse side-effects. To achieve this, there is an urgent need for biomarkers that can help diagnose and monitor the disease process. The term “biomarker” encompasses a broad subcategory of medical indications that can be accurately and consistently measured. In childhood TA, we commonly combine biomarkers including demographics, clinical features, laboratory findings, genetic information, and imaging findings to facilitate early diagnosis and the subsequent monitoring of disease flares.

Traditional inflammatory markers like C-reactive Protein (CRP) and erythrocyte sedimentation rate (ESR) are commonly tested and reported at diagnosis, yet their sensitivity and specificity in detecting disease activity remain uncertain [4,5,6]. Abnormalities in complete blood count (CBC) parameters such as anemia or thrombocytosis are commonly observed in chronic inflammation but appear to lack sensitivity and specificity. Although novel laboratory markers for diagnosing TA and monitoring disease activity are currently undergoing extensive investigation, they have not yet been integrated into routine clinical practice [7,8]. Additionally, imaging plays a pivotal role in childhood-onset TA, not only for diagnosis and assessing disease activity but also for guiding management and clinical care [9,10,11]. Lifelong follow-up necessitates noninvasive imaging techniques with minimal radiation exposure.

Limited biomarker data exist in childhood TA due to low patient numbers and lack a uniform approach, making it difficult to generalize the existing data. The aim of this review was to characterize the clinical, laboratory, and imaging presentation of childhood-onset TA, and to identify and classify potential biomarkers that have been investigated and reported in the literature to date. By consolidating current knowledge, this review aims to inform future prospective research, ultimately enhancing patient care and outcomes in childhood-onset TA.

## 2. Materials and Methods

The systematic review was conducted and reported in accordance with the Preferred Reporting Items for Systematic reviews and Meta-Analyses guidelines (PRISMA 2020) [12].

### 2.1. Search Strategy

An experienced academic librarian conducted an electronic, language-unrestricted search from each database’s inception to 15 August 2023. Databases included Ovid MEDLINE, Ovid EMBASE, Wiley Cochrane Library, ClinicalTrias.gov, and WHO ICTRP. Reference lists of relevant papers were manually screened for additional studies. Search terms were used to identify all articles related to childhood-onset TA. Our search was limited to the pediatric age group (0 to 18 years) and included all studies published after 2000 (extending our research period due to the rarity of TA). The detailed search strategy is presented in the Appendix A.

### 2.2. Selection Criteria

The review included original prospective and retrospective studies that reported data on the diagnosis of Takayasu’s arteritis in children. Given the rarity of this disease in childhood, the majority of our data were extracted from observational studies and case series. We excluded duplicated studies, case series with fewer than 3 cases, reviews, conference abstracts, posters, letters, and articles focusing solely on the treatment of TA. Additionally, we excluded articles on TA in adults or mixed cohorts if the extraction of data specifically related to children was not feasible. Articles written in languages other than English were also excluded from our analysis. Refer to the PRISMA flow diagram for our study selection process (Figure 1).

### 2.3. Data Extraction

The Covidence data management software (www.covidence.org) was used to screen and review eligible articles. Title and abstract screening were performed independently by three authors (LP, LH, and MT). Studies selected for full-text screening were further evaluated for eligibility based on inclusion and exclusion criteria, with this process independently performed by the same three investigators. Conflicts were resolved through discussion between investigators. Data extraction was performed by LP using a standardized form through the Covidence program, after which MT checked the data for correctness and completeness. Descriptive data extracted from each study included demographic data, clinical features, laboratory measurements, diagnostic imaging, and genetic analysis (if performed).

### 2.4. Quality Assessment

The quality of observational studies and case series was assessed using the Oxford 2011 Levels of Evidence. LP conducted the assessment with verification by MT [13]. Discrepancies were resolved through discussion between investigators.

### 2.5. Statistical Analysis

Data were captured in a designated research database. Descriptive analyses were performed in Excel (Version 6.86) and R (Version 2023.12.0+369). For most variables, we used the sum of cases from studies reporting on the variable. Consequently, the denominator in the proportions depended on the studies reporting on a given variable.

## 3. Results

### 3.1. Study Characteristics

#### 3.1.1. Included Studies

The PRISMA flow diagram depicts the different phases of this systematic review (Figure 1). We identified 2026 articles published between 2000 and 2023 that met our inclusion criteria. After removing duplicates, 1957 unique publications were screened.

#### 3.1.2. Characteristics of Included Studies

We excluded 1605 articles based on title and abstract screening. Following full-text screening, an additional 300 articles were excluded for the following reasons: mixed cohort (*n* = 154), fewer than three pediatric cases (*n* = 39), adult population (*n* = 27), non-English language (*n* = 20), conference abstract/poster/letter (*n* = 20), topic focused on treatment of TA (*n* = 19), overlapping cohorts (*n* = 9), case report with fewer than three cases (*n* = 5), topic not focusing on TA (*n* = 3), review (*n* = 2), wrong outcome (*n* = 1), and protocol/trial design (*n* = 1).

All studies included for analysis (*n* = 42) were non-controlled case series published after 2000, mostly retrospective (95%) [14,15,16,17,18,19,20,21,22,23,24,25,26,27,28,29,30,31,32,33,34,35,36,37,38,39,40,41,42,43,44,45,46,47,48,49,50,51,52,53,54,55]. These studies were primarily conducted in China (*n* = 11), India (*n* = 8), Turkey (*n* = 6), and the USA (*n* = 5). The remaining studies (*n* = 10) did not contain extractable data and were therefore not included in the analysis [56,57,58,59,60,61,62,63,64,65].

#### 3.1.3. Quality Assessment

The quality of all included studies met the standards of the Oxford 2011 Levels of Evidence [13]. As all articles were case series, they qualified for Level 4 evidence.

### 3.2. Demographics

In total, 1067 pediatric patients diagnosed with TA were reported across 42 original publications. The diagnosis was based on the most recent EULAR/PRINTO/PRES criteria [3] in 52.4% of studies (or 79% of studies performed after publication of EULAR/PRINTO/PRES criteria) [15,17,19,20,21,23,24,29,30,37,38,39,40,41,45,46,47,48,49,51,53,55], while others [14,16,25,26,28,31,32,33,34,35,43] used ACR criteria [66] and/or EULAR/PRES criteria [67]. The EULAR/PRINTO/PRES criteria mandate angiographic abnormalities of the aorta or its main branches, in combination with at least one of the following features: (a) decreased peripheral artery pulse(s) and/or claudication of extremities, (b) blood pressure difference >10 mmHg, (c) bruits over the aorta and/or its major branches, (d) hypertension (related to childhood normative data), and (e) increased acute-phase reactants (ESR above 20 mm/h or CRP above normal). The ACR criteria require at least three of the following criteria: (a) age at disease onset ≤40 y, (b) claudication of extremities, (c) decreased brachial artery pulse, (d) blood pressure difference >10 mmHg, (e) bruits over subclavian arteries or aorta, and (f) angiographic abnormalities. The EULAR/PRES criteria necessitate angiographic abnormalities of the aorta or its main branches, in combination with at least one of the following features: (a) decreased peripheral artery pulse(s) and/or claudication of extremities, (b) blood pressure difference >10 mm Hg, (c) bruits over aorta and/or its major branches, and (d) hypertension (related to childhood normative data).

Figure 2 illustrates the distribution of mean or median age at symptom onset (*n* = 502) and at diagnosis (*n* = 464). The mean or median age could not be calculated, due to the absence of individual patient data. However, Figure 2 shows an age distribution between 10 and 15 years, with cases occurring across all ages, even as early as 1.5 months. A significant delay between symptom onset and diagnosis was observed, ranging from 0 to 20 months (*n* = 676) with a maximum delay of 264 months (Figure 2). Additionally, there was a predominance of females among childhood-onset TA, with female patients accounting for 72% (*n* = 1038) [14,15,16,17,18,19,20,21,22,23,24,25,26,27,28,29,30,31,32,34,35,37,38,40,41,42,43,44,45,46,47,48,49,50,51,52,53,54,55].

### 3.3. Clinical Features

Clinical features in childhood-onset TA were categorized into (a) general symptoms, (b) cardiovascular symptoms, (c) neurological symptoms, (d) musculoskeletal symptoms, (e) respiratory symptoms, (f) gastrointestinal symptoms, and (g) other symptoms (renal, ocular, and dermatological involvement). General, cardiovascular, and neurological symptoms were predominantly noted at presentation in childhood-onset TA (Figure 3).

#### 3.3.1. General Symptoms [14,15,16,17,18,19,20,21,22,23,24,25,26,27,28,29,30,31,32,35,37,38,39,40,41,42,43,44,45,46,47,48,51,53,54]

Constitutional findings (fever, asthenia, malaise, weight loss) were commonly reported at presentation (64%, *n* = 109). When recorded separately, fever was most frequent (36%, *n* = 593), followed by malaise/fatigue (34%, *n* = 369), weight loss (25%, *n* = 372), and anorexia (25%, *n* = 24). Night sweats (9%, *n* = 55) and lymphadenopathy (9%, *n* = 44) were less common.

#### 3.3.2. Cardiovascular Symptoms [14,15,16,17,18,19,20,21,22,23,24,25,26,27,28,29,30,31,32,35,36,37,38,39,40,41,42,43,44,45,46,47,48,50,51,52,53,54,55]

Hypertension was the most frequent symptom at presentation (70%, *n* = 731), followed by decreased or absent pulses (53%, *n* = 699), blood pressure discrepancy (51%, *n* = 394), bruits (45%, *n* = 753), pulse inequality (34%, *n* = 64), claudication (31%, *n* = 583), and heart failure (30%, *n* = 373). In some case series, hypertension was divided into systolic hypertension (66%, *n* = 129) and diastolic hypertension (44%, *n* = 129). Less common symptoms included palpitations (20%, *n* = 143), cardiomyopathy (19%, *n* = 205), chest pain (12%, *n* = 366), myocardial ischemia (7%, *n* = 141), carotidynia (5%, *n* = 214), and gangrene (4%, *n* = 84).

#### 3.3.3. Neurological Symptoms [14,16,17,18,20,21,22,23,24,25,27,28,29,32,35,37,38,39,40,41,44,45,46,47,48,51,52,53,55]

Headache (35%, *n* = 510) and dizziness (29%, *n* = 216) were the most frequent symptom at presentation. Some case series reported on neurological symptoms defined as headache, stroke, or syncope, noted in 70% of cases (*n* = 71). Additional symptoms included seizures (19%, *n* = 198), hemiparesis (19%, *n* = 27), syncope (11%, *n* = 469), stroke or transient ischemic attack (TIA) (10%, *n* = 456), and paresthesias (7%, *n* = 28).

#### 3.3.4. Musculoskeletal Symptoms [14,16,17,18,20,21,22,23,25,27,28,32,37,38,39,40,41,43,44,46,47,50,51,52,54,55]

Musculoskeletal symptoms such as arthritis, arthralgia, myalgia, and limb pain, though less typical compared to other rheumatological diseases, were reported in 29% of cases (*n* = 204). Back pain (20%, *n* = 61), arthritis/arthralgia (15%, *n* = 337), and myalgia (14%, *n* = 43) were the most commonly noted.

#### 3.3.5. Respiratory Symptoms [14,16,19,20,22,23,24,27,28,36,37,38,40,41,42,46,48,51,55]

Dyspnea (31%, *n* = 537) was the most frequent respiratory symptom at presentation, followed by respiratory failure (27%, *n* = 11), infiltrate (21%, *n* = 14), cough (17%, *n* = 24), and pleural effusion (13%, *n* = 45).

#### 3.3.6. Gastro-Intestinal Symptoms [14,22,24,25,27,28,30,32,35,37,39,42,43,44,45,46,48,50,51,52,53]

Nausea/vomiting (18%, *n* = 209) and abdominal pain (15%, *n* = 343) were the most frequently reported gastrointestinal symptom at presentation.

#### 3.3.7. Other Symptoms [14,16,18,19,20,22,24,25,27,32,35,37,38,39,40,41,44,45,46,47,48,51,52,55]

Ocular involvement (14%, *n* = 655), renal involvement (17%, *n* = 265), and dermatological involvement (18%, *n* = 77) were also reported. Ocular involvement included visual changes or disturbances, visual loss, uveitis, retinal hemorrhage, and blindness. Renal involvement included proteinuria, hematuria, abnormal urine output, and elevated serum creatinine. Dermatological involvement included pyoderma gangrenosum-like vasculitic ulceration and other rashes.

### 3.4. Laboratory Features

Figure 4 illustrates the distribution of CRP (mean or median, *n* = 267) and ESR (mean or median, *n* = 347) in included pediatric case series, showing elevated levels in most patients. CRP ranged predominantly between 1 and 20 mg/dL, and ESR ranged between 30 and 100 mm/h. Some case series only indicated normal or elevated levels of these markers, with 51% reporting elevated CRP (*n* = 344), 53% elevated ESR (*n* = 463), and 68% elevated CRP or ESR (*n* = 198) [14,16,17,18,19,20,21,22,23,24,26,27,28,35,37,38,40,41,42,43,44,45,47,48,50,51,53,55].

A limited number of studies reported CBC results, showing extensive heterogeneity without consistent abnormalities. Some case series only reported on low hemoglobin (43%, *n* = 162), elevated WBC count (61%, *n* = 75), or elevated platelet count (44%, *n* = 70) [14,16,19,22,24,28,41,42,45].

Additional laboratory markers were reported with low patient numbers, including impaired renal function noted in 13% (*n* = 204) [14,22,35,38,48,51]. Positive tuberculosis (TB) screening was seen in 23% (*n* = 288) (screening performed by PPD skin test, sputum testing, or IFN-g release test), with the highest positive TB results in Brazil, India, and China [14,18,22,24,25,27,36,40,41,42,45,51,52]. Incidental reports included elevated troponin levels in 4 out of 14 patients [17,46], elevated renin in 5 out of 5 patients [45], positive autoantibodies (not specified) in 4 out of 15 patients [45,46], positive antinuclear antibodies (ANA) in 11 out of 31 patients [42,46,51], and positive antineutrophil cytoplasmic antibodies (ANCA) in in 2 out of 29 patients [45,46,51]. Genetic testing was not performed in the studies included.

### 3.5. Imaging Features

Common imaging modalities for diagnosing childhood-onset TA included conventional angiography (76%), MRI/MRA (25%), CT/CTA (10%), and FDG-PET (1%) (*n* = 403) [14,16,17,18,19,20,22,23,24,25,27,28,33,35,38,39,40,41,42,43,44,46,51,52,53].

The angiographic classification in childhood-onset TA (Figure 5) primarily utilized the Numano classification [68] (*n* = 636) [14,16,20,21,22,29,30,31,32,33,34,40,41,45,46,47,48,49], followed by the Lupi–Herrera classification [69] (*n* = 130) [15,25,27,38] and Pantonowitz classification (*n* = 55) [52]. Studies using the Numano classification showed type V (53%) in more than half of the patients, followed by type IV (21%), type I (8%), type III (8%), type IIa (4%), and type IIb (4%). Studies using the Lupi–Herrera classification most frequently showed type III (35%), followed by type II (31%), type I (30%), and type IV (4%).

Lesions predominantly involved narrowing/stenosis (53%) and occlusion (37%), with aneurysm (10%), wall thickening (9%), dilation (5%), and inflammation (1%) less frequently observed (*n* = 297) [17,18,19,23,24,28,29,35,40,41,42,43,46,48].

The abdominal aorta (54%) and renal arteries (44%) emerged as the most commonly affected vessels in childhood-onset TA, followed by the thoracic aorta (33%), subclavian arteries (32%), and carotid arteries (30%) according to a sample size of 712 patients [14,17,18,19,20,21,22,23,24,25,26,27,28,29,30,32,33,34,35,36,37,38,39,41,42,43,45,48,49,51,53]. Figure 6 illustrates the distribution of the most frequently affected vessels in childhood-onset TA (>5% of patients), with the potential involvement of any artery in the body (refer to Appendix A for a comprehensive list of all affected arteries reported).

### 3.6. Additional Studies on Imaging Approaches

The following studies met our inclusion criteria but focused on different outcomes compared to previous case series, so they are discussed separately.

Grotenhuis et al. utilized a combination of modalities including MRI, vascular ultrasound, applanation tonometry, and echocardiography to demonstrate increased arterial stiffness of the aorta and its branches in children with TA. This stiffness correlated with increased PVAS scores and hypertension [60]. Additionally, the team observed an increased LV mass index, altered myocardial deformation, and impaired diastolic function. Integrating these imaging modalities could provide an opportunity to assess and follow the cardiovascular health in children with TA.

Nozawa et al. retrospectively collected pediatric patients with fever of unknown origin who were diagnosed with TA using ultrasonography [58]. Ultrasonography detected abnormal arteries in the abdomen and neck regions with a moderate concordance rate compared to CT, suggesting the utility of ultrasonography as a screening tool in childhood-onset TA.

Sun et al. utilized CTA with Deep Learning Image Reconstruction (DLIR) in children with TA, noting improved image quality and reduced radiation dose [56]. DLIR algorithms preserved image detail while minimizing noise, crucial for radiation safety in pediatric patients.

Clemente et al. explored the potential of PET/MRA in children with TA, revealing that 18F-FDG-PET identified additional positive findings compared to MRA alone [61]. Furthermore, the FDG-PET/MRA findings of this cohort were compared with imaging in age- and sex-compared oncology patients, showing significant 18F-FDG uptake in the vessel wall of large vessels in most TA patients in comparison to the absence of evident arterial uptake in the oncology patients [62]. Despite advancements in FDG-PET/MRA imaging, uncertainties persist regarding the implications of increased vascular FDG uptake and its correlation with TA disease activity. Additionally, the absence of a gold standard imaging technique for detecting TA disease activity complicates the validation of imaging parameters. However, these findings suggest that FDG-PET/MRA could serve as a potential marker of disease activity in childhood-onset TA, offering valuable insights to guide treatment decisions.

### 3.7. Disease Activity Scores

EULAR recommends the Birmingham Vasculitis Activity Score (BVAS) and the Disease Extent Index (DEI) to iteratively determine disease activity in adults with systemic chronic vasculitis [70,71,72]. These tools are currently not systematically used in childhood vasculitis studies. Demirkaya et al. demonstrated that BVAS and DEI have adequate convergent validity and the ability to differentiate among different types of childhood vasculitis [63]. Subsequently, Dolezalova et al. proposed the Pediatric Vasculitis Activity Score (PVAS) as a uniform tool for assessing disease activity in childhood vasculitis, offering potential utility in childhood-onset TA management [73].

## 4. Discussion

To our knowledge, this systematic review represents the first comprehensive effort to collect and analyze all available data on the diagnosis of childhood-onset TA. With a total of 1067 patients included, our analysis provides a robust overview of the presenting features of childhood-onset TA. Our review aimed to identify and classify potential biomarkers for childhood-onset TA and to develop practical guidelines for health care providers, as prompt and accurate diagnosis is crucial for ensuring optimal treatment leading to improved outcomes. The diverse and heterogeneous presentation of childhood-onset TA presents significant challenges for timely diagnosis. Unfortunately, there is a paucity of data on biomarkers in childhood-onset TA, and currently, no single biomarker exists for diagnosis. Given the varied clinical presentation, the lack of specific and sensitive laboratory markers, and the diagnostic challenges associated with imaging techniques, a combination of biomarkers is required for an accurate diagnosis.

A female predominance was clearly present in childhood-onset TA, although less pronounced than in adults [74]. The cause of this sex difference remains unknown, and to our knowledge, the underlying causal mechanisms have not yet been investigated in TA. The age distribution at symptom onset and diagnosis was situated between 10 and 15 years, although the condition can manifest at any age. There is a known delay between symptom onset and diagnosis, ranging between 0 and 20 months, similar to observations in adult-onset TA.

Our review confirmed the heterogeneous clinical spectrum of childhood-onset TA (Figure 7). Childhood-onset TA presented predominantly with cardiovascular symptoms, general symptoms, and neurological symptoms. Most frequent cardiovascular symptoms included hypertension, decreased or absent pulses, and blood pressure discrepancy. These findings correspond with our observation of a predominant vascular disease pattern affecting the abdominal aorta and renal arteries in children. This also explains the differing symptomatology between adult-onset and childhood-onset TA, as adults appear to have more involvement of the supradiaphragmatic large vessels [74]. General symptoms such as fever and malaise, while nonspecific in children, necessitate a broad differential diagnosis including TA. Similarly, headache and dizziness are also nonspecific and are probably linked with the high rates of hypertension. In our review, percentages of clinical symptoms are difficult to interpret, as not all included studies reported on every single symptom, leading to a significant bias. Nevertheless, these numbers still outline the general trends in childhood-onset TA. The diagnostic challenge of childhood-onset TA persists due to its diverse clinical presentation, warranting the consideration of infectious diseases, (auto-)inflammatory diseases, cardiovascular diseases, fibromuscular dysplasia, and genetic syndromes, depending on the presenting symptoms.

Laboratory findings lack sensitivity and specificity for diagnosing childhood-onset TA. Elevated levels of CRP and ESR, for example, were commonly observed, with ESR often higher than CRP in many cases, although normal values cannot definitively rule out disease. There were no consistent complete blood count abnormalities found, even though abnormalities of these values are generally noted in chronic inflammatory diseases. Despite the renal arteries being one of the most affected arteries, only 13% showed renal impairment. Interestingly, TB screening was positive in 23% of cases, although only conducted in 288 children, suggesting that this may be an overestimation. The highest positive TB results were found in countries with a high TB disease burden [75]. Even though the exact connection between (latent) TB infection and childhood-onset TA is still unclear, the high prevalence of positive TB screening, especially in endemic areas, supports the importance of screening and treating (latent) TB infection in children with TA. Autoantibodies were rarely reported and were mainly used to exclude other rheumatological diseases. Although the pathogenesis of TA likely involves complex genetic predisposition, genetic testing in childhood-onset TA was not reported in the included studies.

Imaging modalities for childhood-onset TA vary among countries based on resources and local guidelines, with conventional angiography being the most widely used. While it was once the gold standard for TA diagnosis, its high radiation and invasiveness have made it less favored, particularly in children. EULAR recommendations for imaging in large-vessel vasculitis suggest MRA as the primary diagnostic imaging modality for TA, although these guidelines are not specific to pediatric patients [10]. CTA and FDG-PET are alternative options for suspected TA cases.

Numerous emerging biomarkers are currently under investigation for both the early diagnosis of TA and for assessing disease activity [7,8]. For instance, various cytokines, such as chemokine, tumor-necrosis factor-alpha (TNF-α), and interferon-gamma (IFN-γ), interleukin (IL)-6, 8, 10, 12, 18, and 23, show promise for diagnosing TA or might serve as useful markers for assessing disease activity [76,77,78,79]. Several matrix metalloproteinases (MMPs) (1, 2, 3, 9), which are induced by cytokines, and their tissue inhibitors (TIMPs) may contribute to vascular injury and remodeling in TA [80,81,82,83]. Pentraxin-3 (PTX3), an acute-phase protein produced by various tissues and cells, has also been the focus of numerous studies. These studies have reported elevated PTX-3 levels in TA patients compared to healthy controls, and additionally, PTX-3 has been linked to the active stage of TA [83,84,85]. There have been some promising reports of elevated plasma N-terminal pro-brain natriuretic peptide (NT-proBNP) levels in adult patients with active TA compared to those with inactive disease and to healthy controls [86,87]. Additionally, extensive arterial involvement as well as inflammatory markers (CRP, ESR, WBC count) appear to be associated with NT-proBNP levels in TA. Alongside several other biomarkers, these examples hold promise for offering crucial insights into early TA diagnosis, disease activity assessment, and treatment guidance. However, despite these advancements, no biomarkers have yet been established for reliable clinical use, and the current focus has primarily been on adults. Additionally, there were many inconsistencies concerning biomarkers in different cohorts, suggesting unreliability in using a single biomarker in TA. This might necessitate the development of a multiple-biomarker-based model for diagnosis and disease activity classification [88].

Our review has several limitations that warrant acknowledgement. Firstly, all included studies were retrospective case series, inherently associated with the limitations of this study design. Additionally, it did not involve collecting individual patient data, potentially introducing publication bias and limiting the evidence available for analysis. Furthermore, the relatively small total numbers of patients across studies may have introduced bias, as not all studies reported on every element of interest. While we aimed to enhance representativeness by excluding case reports with fewer than three cases, the rarity of TA may have led to the exclusion of potentially informative clinical cases. Lastly, the management and prognosis of childhood-onset TA are additionally challenging topics but were outside the scope of this review. Randomized controlled clinical trials have not been performed for childhood-onset TA, primarily due to the low number of patients. Therapeutic recommendations are primarily based on observational data and extrapolated from adult studies but also vary significantly between different regions of the world. Additionally, comparing the prognosis between studies is extremely difficult, as there are no universal definitions for disease activity or remission, complicating the description and comparison of prognostic data.

Future clinical trials focusing on childhood-onset TA face significant challenges due to the rarity of the disease. Compounding this issue is the existence of various classification criteria, which complicates data comparison in pediatric TA studies. The EULAR/PRINTO/PRES classification criteria stand out as the most recent and comprehensive criteria, incorporating modern imaging modalities, acute-phase reactants, and pediatric-specific symptoms. In view of the demonstrated high sensitivity and specificity, it is encouraged to uniformly adopt this classification in future studies to facilitate enhanced data comparability. Multicentric collaborative efforts will be indispensable for accumulating sufficient standardized data. Such collaborations are vital not only for gaining a better understanding of the natural course of childhood-onset TA, but also for identifying sensitive and specific biomarkers for diagnosis and disease activity. Ultimately, these collective efforts will undoubtedly contribute to improved disease outcomes in children diagnosed with TA.

## Figures and Tables

**Figure 1 ijms-25-07345-f001:**
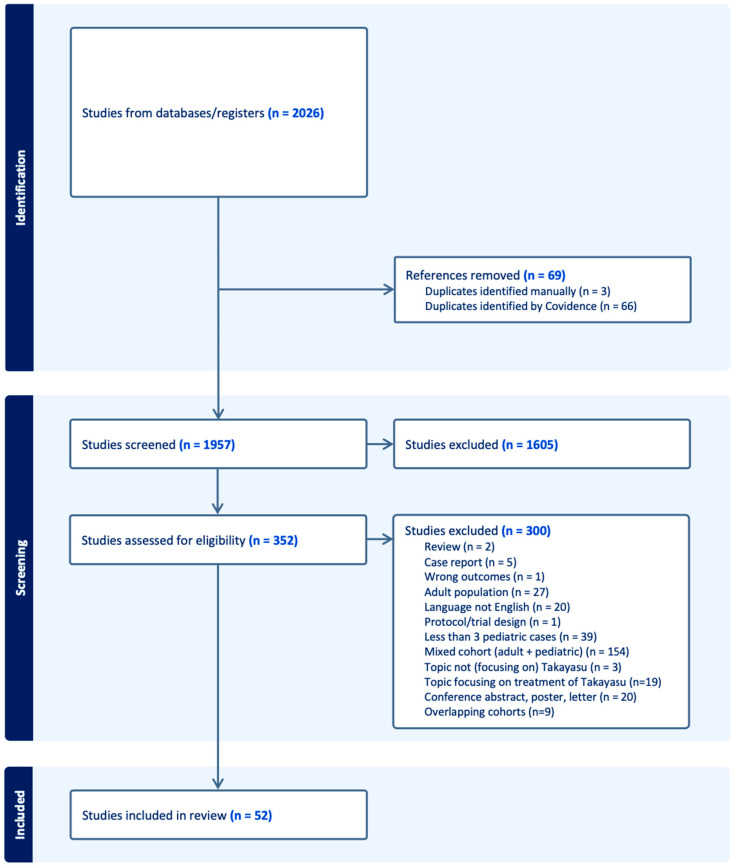
Preferred Reporting Items for Systematic Reviews and Meta-Analyses (PRISMA) flow diagram.

**Figure 2 ijms-25-07345-f002:**
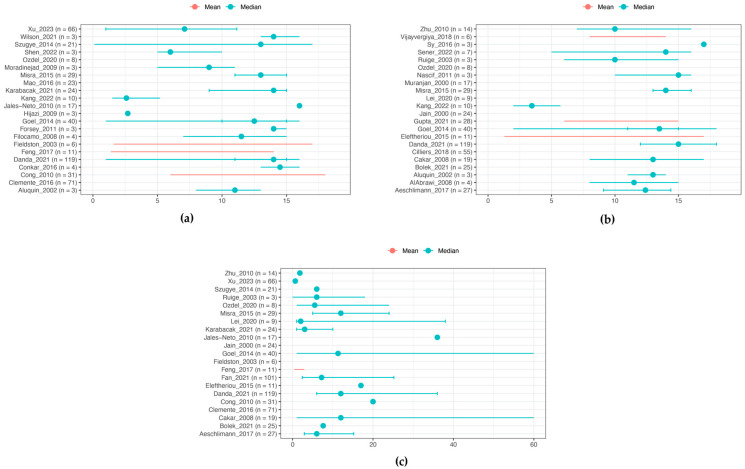
Demographics at presentation. (**a**) Age at symptom onset (years) [15,16,17,18,19,21,23,26,28,29,30,32,35,37,40,41,42,43,44,45,47,48]; (**b**) age at diagnosis (years) [14,18,20,21,22,24,25,27,30,31,33,38,40,46,47,48,49,50,51,52,53,54]; (**c**) time between symptom onset and diagnosis (months) [14,15,16,21,22,25,29,31,32,33,37,40,41,44,45,46,47,48,49,51,55]. Mean values are displayed in red, and median values are displayed in blue. Vertical lines represent the interquartile range (IQR), while the beginning and ending points of the horizontal lines indicate the minimum and maximum values.

**Figure 3 ijms-25-07345-f003:**
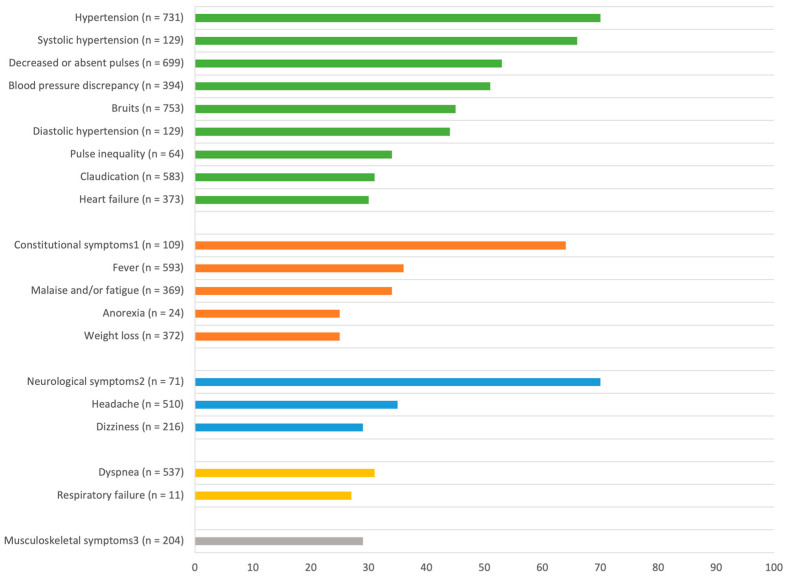
Most frequently (≥25%) described symptoms at presentation, divided according to different subcategories. Green: cardiovascular symptoms; orange: general symptoms; blue: neurological symptoms; yellow: respiratory symptoms; grey: musculoskeletal symptoms.

**Figure 4 ijms-25-07345-f004:**
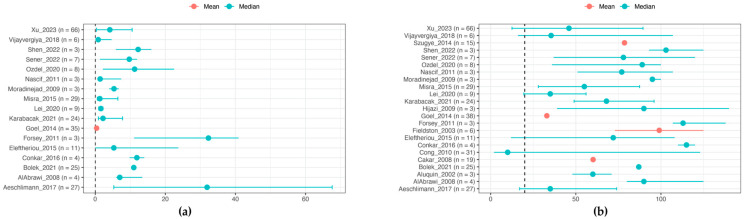
Inflammatory parameters at presentation. (**a**) CRP (mg/dL) [15,19,20,21,23,24,28,29,31,40,42,46,47,49,50,51,53]; (**b**) ESR (mm/h) [15,16,18,19,20,21,23,24,25,28,29,31,35,37,40,42,44,46,47,49,50,51,53]. Mean values are displayed in red and median values are displayed in blue. Vertical lines represent the interquartile range (IQR), while the beginning and ending points of the horizontal lines indicate the minimum and maximum values. The dashed line represents the upper limit of normal (CRP 0.5 mg/dL, ESR 20 mm/h).

**Figure 5 ijms-25-07345-f005:**
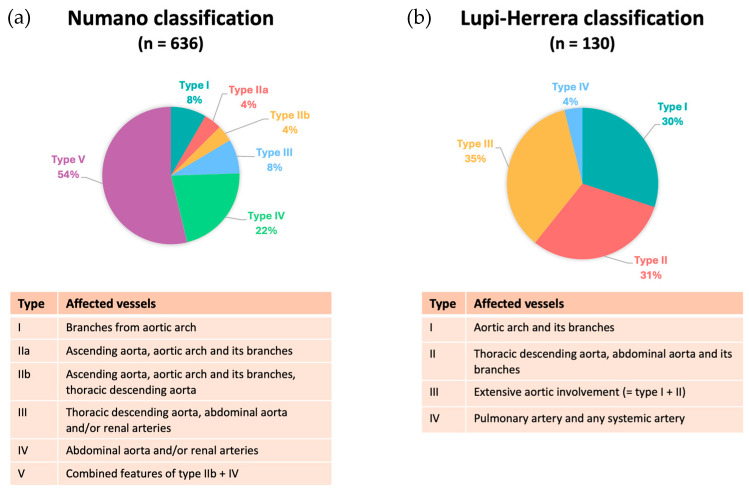
Angiographic classification of affected vessels. (**a**) Numano classification; (**b**) Lupi–Herrera classification.

**Figure 6 ijms-25-07345-f006:**
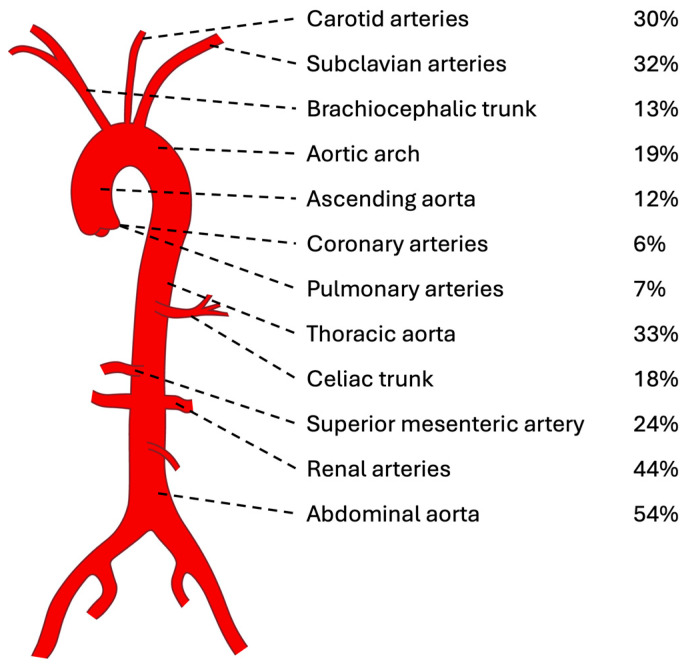
Most frequently (>5%) involved arteries in childhood-onset TA.

**Figure 7 ijms-25-07345-f007:**
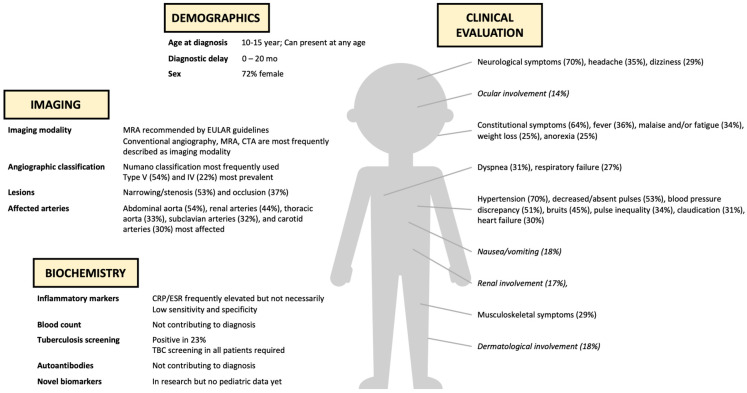
Summary of diagnostic biomarkers in childhood-onset TA based on current systematic review. Clinical evaluation includes most frequently (≥25%) described symptoms at presentation (ocular, gastro-intestinal, renal, and dermatological involvement (italic) is described but less common (<25%).

## Data Availability

Not applicable.

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
