# Peer review of "Real-World Biomarkers for Pediatric Takayasu Arteritis"

_ijms, 2024, doi:10.3390/ijms25137345_

Round 1
Reviewer 1 Report
Comments and Suggestions for Authors
The Authors performed a systematic literature review to identify and classify all candidates for biomarkers of childhood-onset Takayasu Arteritis (TA). A total of 1067 TA patients were included with a predominant involvement of cardiovascular, constitutional, and neurological symptoms. Laboratory parameters exhibited low sensitivity and specificity. Imaging predominantly revealed involvement of the abdominal aorta and renal arteries, with magnetic resonance angiography (MRA) being the preferred imaging modality. The manuscript is well written, but please find below some comments:
-in the paragraph on serum biomarkers, I wonder whether the Authors found something about natriuretic peptides and troponin
-What about medical therapy? Were there any data about patients' management?
-What about prognosis? Were there any data about patients's outcome and follow-up?
Comments on the Quality of English LanguageFine
Author Response
Dear Reviewer 1,
We would like to thank the reviewer for their critical review of our manuscript. Please find below our response and the subsequent changes we have made in the manuscript.
The Authors performed a systematic literature review to identify and classify all candidates for biomarkers of childhood-onset Takayasu Arteritis (TA). A total of 1067 TA patients were included with a predominant involvement of cardiovascular, constitutional, and neurological symptoms. Laboratory parameters exhibited low sensitivity and specificity. Imaging predominantly revealed involvement of the abdominal aorta and renal arteries, with magnetic resonance angiography (MRA) being the preferred imaging modality. The manuscript is well written, but please find below some comments:
Comment 1: In the paragraph on serum biomarkers, I wonder whether the Authors found something about natriuretic peptides and troponin
Response 1: We want to thank the reviewer for the additional biomarker information. The included case series in our review on childhood-onset TA did not report anything concerning natriuretic peptides. However, in adults, there have been some promising reports of elevated plasma N-terminal pro-brain natriuretic peptide (NT-proBNP) levels in patients with active TA compared to those with inactive disease and to healthy controls. Patients with TA exhibiting extensive arterial involvement or complications also showed significantly higher NT-proBNP levels. Finally, inflammatory markers such as CRP, ESR, and WBC count were independently associated with NT-proBNP levels in TA. Additionally, we identified two case series in our review that reported troponin results [17, 46] (row 262). The data showed elevated troponin levels in 2 out of 11 patients [46], and in 2 out of 3 patients (all presenting with acute myocardial infarction) [17]. Unfortunately, due to the infrequency of these reports, it is challenging to draw definitive conclusions. For now, these findings seem particularly notable in cases presenting with ischemic cardiac symptoms.
In light of this interesting remark, we included the following statement in the manuscript (page 11, row 405):
“There have been some promising reports of elevated plasma N-terminal pro-brain natriuretic peptide (NT-proBNP) levels in adult patients with active TA compared to those with inactive disease and to healthy controls. Additionally, extensive arterial involvement as well as inflammatory markers (CRP, ESR, and WBC count) appear to be associated with NT-proBNP levels in TA.“
Comment 2 and 3: What about medical therapy? Were there any data about patients' management? What about prognosis? Were there any data about patient’s outcome and follow-up?
Response 2 and 3: We agree with reviewer 1 that medical treatment for Takayasu arteritis is of interest. This would be a topic for a separate review. We opted to not include any treatment as not to confuse the readers as we will not be able to provide all the evidence within this review on real-world biomarkers. Identical to the question on treatment, we agree patient outcome and follow-up would be of interest but would fit better in a review on treatment and not within the scope of a review on biomarkers. RCTs (randomized controlled clinical trials) have not been performed within childhood-onset TA, primarily due to the low number of patients and the rarity of the disease. Therapeutic recommendations are primarily based on observational data and extrapolated from adult studies but also vary significantly between different regions of the world. Additionally, comparing prognosis between studies is extremely difficult, as there are no universal definitions for disease activity or remission, complicating the description and comparison of prognostic data.
Considering this valid remark, we included the following statements (page 11, row 424):
“Lastly, management and prognosis of childhood-onset TA are additionally challenging topics but were outside the scope of this review. Randomized controlled clinical trials have not been performed for childhood-onset TA, primarily due to the low number of patients. Therapeutic recommendations are primarily based on observational data and extrapo-lated from adult studies but also vary significantly between different regions of the world. Additionally, comparing prognosis between studies is extremely difficult, as there are no universal definitions for disease activity or remission, complicating the description and comparison of prognostic data.”
Reviewer 2 Report
Comments and Suggestions for Authors
The present article tries to find the answer to an important problem related to Takayasu's arteritis, namely, the identification of some biomarkers in the pediatric population.
The article is interesting and useful, it evaluated the existing data in the literature using currently accepted and validated methods.
Some suggestions to ease the read
- Some of included data should be sustained by references – see rows 67-69 (lack of specificity and sensitivity of CRP, row 74 pivotal role of imaging, row 319 EULAR recommendations, BVAS, DEI
- Figure 1, which is very important related to the study design, should be rearranged so that it is easier to read
- Row 157 – since you described EULAR/PRINTO/PRES you should described all 3 criteria used. Also, in order to be easier to read, you can include all 3 criteria in one table, each on one column and associate related references (studies trat included that particular criteria)
- Chapter 3.3, clinical features should be more condensed, your one phrase paragraphs are a little bit peculiar
- Chapter 3.4 Laboratory features – did you find any correlations between inflammatory markers and age, clinical manifestations, area involved etc?
- Regarding positive TB screen have you checked the area where the patients come from? Is there any provenance from endemic areas?
- For figure 7, the writing is small compared to the text and is difficult to read
Author Response
Dear Reviewer 2,
We would like to thank the reviewer for their critical review of our manuscript. Please find below our response and the subsequent changes we have made in the manuscript.
The present article tries to find the answer to an important problem related to Takayasu's arteritis, namely, the identification of some biomarkers in the pediatric population. The article is interesting and useful, it evaluated the existing data in the literature using currently accepted and validated methods.
Some suggestions to ease the read:
Comment 1: Some of included data should be sustained by references – see rows 67-69 (lack of specificity and sensitivity of CRP, row 74 pivotal role of imaging, row 319 EULAR recommendations, BVAS, DEI
Response 1: We thank the reviewer for identifying the lack of supporting literature for some of the overall statements in the manuscript. We have added the appropriate references as requested:
=> Rows 67-69 (lack of specificity and sensitivity of CRP:
We have added reference [4-6] to cite this statement.
=> Row 74 (-> 76) pivotal role of imaging:
We have added reference [9-11] to cite this statement.
=> Row 319 (-> 330) EULAR recommendations, BVAS, DEI:
We have added reference [70-72] to cite the recommendations, BVAS and DEI.
Comment 2: Figure 1, which is very important related to the study design, should be rearranged so that it is easier to read
Response 2: We acknowledge that not everyone might be as familiar with the PRISMA flow diagram, and it might not be an easy overview. However, this is the most common used flow diagram to picture the study design, so we do propose to keep it in its current form as depicted in many reviews.
Comment 3: Row 157, since you described EULAR/PRINTO/PRES you should described all 3 criteria used. Also, in order to be easier to read, you can include all 3 criteria in one table, each on one column and associate related references (studies trat included that particular criteria)
Response 3: We thank the reviewer for identifying the lack of our explanation of all three potential criteria used and we have added the definitions in the manuscript.
(Page 5, row 164):
“The ACR criteria require at least three of the following six criteria: age at disease onset ≤ 40y, claudication of extremities, decreased brachial artery pulse, blood pressure difference >10mmHg, bruits over subclavian arteries or aorta, and angiographic abnormalities. The EULAR/PRES criteria necessitate angiographic abnormalities of the aorta or its main branches, in combination with at least one of the following four features: decreased peripheral artery pulse(s) and/or claudication of extremities, blood pressure difference >10 mm Hg, bruits over aorta and/or its major branches, and hypertension (related to childhood normative data).”
Comment 4: Chapter 3.3, clinical features should be more condensed, your one phrase paragraphs are a little bit peculiar
Response 4: We acknowledge reviewer 2’s comments on the shorter paragraphs. We do believe these separate paragraphs are important for clinicians who want to focus on a certain organ system and associated clinical features and the literature to have a closer look if needed.
Comment 5: Chapter 3.4, laboratory features – did you find any correlations between inflammatory markers and age, clinical manifestations, area involved etc?
Response 5: Reviewer 2 raises an interesting question concerning possible correlations. Unfortunately, we were not able to collect the individual patient data of the included studies in our review for a pooled analysis. We agree studying correlations between inflammatory markers and clinical presentation and disease severity would be a very interesting field to explore in future research.
Comment 6: Regarding positive TB screen have you checked the area where the patients come from? Is there any provenance from endemic areas?
Response 6: The highest number of positive TB screen results in our review came from studies reporting of patients in India, Brazil, and China. All three countries are on the WHO list of high TB burden countries. There might be a potential bias due to the higher rate of TB testing in these regions compared to areas with lower TB incidence.
We added the following remarks to complement our manuscript:
(Page 8, row 261):
“Positive tuberculosis (TB) screening was seen in 23% (n = 288) (screening performed by PPD skin test, sputum testing, or IFN-g release test) [8,12,16,18,19,21,30,34–36,39,45,46], with the highest positive TB results in Brazil, India, and China.”
(Page 11, row 380):
“The highest positive TB results were found in countries with a high TB disease burden. Even though the exact connection between (latent) TB infection and childhood-onset TA is still unclear, the high prevalence of positive TB screening, especially in endemic areas, supports the importance of screening and treating (latent) TB infection in children with TA.”
Comment 7: For figure 7, the writing is small compared to the text and is difficult to read
Response 7: We thank the reviewer for indicating the small text. We will defer the potential changes to Figure 7 to the editorial and type-setting team to determine which position would be best for clarity, size and text flow.
Round 2
Reviewer 1 Report
Comments and Suggestions for Authors
The Authors addressed all comments and improved their manuscript. Please note that Figure 1 is not present (only the text can be seen) in the pdf file. Please check.
Author Response
Comment 1: The Authors addressed all comments and improved their manuscript. Please note that Figure 1 is not present (only the text can be seen) in the pdf file. Please check.
Response 1: We want to thank Reviewer 1 for identifying this error. We added Figure 1 to the manuscript as a figure.